# Multiple measures derived from 3D photonic body scans improve predictions of fat and muscle mass in young Swiss men

Roman Sager[1], Sabine Güsewell[2,3], Frank Rühli[2,4], Nicole Bender[2], Kaspar Staub[2,4] *

**1** Medical Faculty, University of Zurich, Zurich, Switzerland, **2** Institute of Evolutionary Medicine, University of Zurich, Zurich, Switzerland, **3** Clinical Trials Unit, Kantonsspital St. Gallen, St. Gallen, Switzerland, **4** Zurich Center for Integrative Human Physiology (ZIHP), University of Zurich, Zurich, Switzerland

☯ These authors contributed equally to this work.
* kaspar.staub@iem.uzh.ch

## Abstract

### Introduction

Digital tools like 3D laser-based photonic scanners, which can assess external anthropometric measurements for population based studies, and predict body composition, are gaining in importance. Here we focus on a) systematic deviation between manually determined and scanned standard measurements, b) differences regarding the strength of association between these standard measurements and body composition, and c) improving these predictions of body composition by considering additional scan measurements.

### Methods

We analysed 104 men aged 19–23. Bioelectrical Impedance Analysis was used to estimate whole body fat mass, visceral fat mass and skeletal muscle mass (SMM). For the 3D body scans, an Anthroscan VITUSbodyscan was used to automatically obtain 90 body shape measurements. Manual anthropometric measurements (height, weight, waist circumference) were also taken.

### Results

Scanned and manually measured height, waist circumference, waist-to-height-ratio, and BMI were strongly correlated (Spearman Rho>0.96), however we also found systematic differences. When these variables were used to predict body fat or muscle mass, explained variation and prediction standard errors were similar between scanned and manual measurements. The univariable predictions performed well for both visceral fat ($r^2$ up to 0.92) and absolute fat mass (AFM, $r^2$ up to 0.87) but not for SMM ($r^2$ up to 0.54). Of the 90 body scanner measures used in the multivariable prediction models, belly circumference and middle hip circumference were the most important predictors of body fat content. Stepwise forward model selection using the AIC criterion showed that the best predictive power ($r^2$ up to 0.99) was achieved with models including 49 scanner measurements.

**Data Availability Statement:** All relevant data are within the manuscript and its Supporting Information files.

**Funding:** Mäxi Foundation, Zurich (Grantee Frank Rühli). The funders had no role in study design, data collection and analysis, decision to publish, or preparation of the manuscript.

**Competing interests:** The authors have declared that no competing interests exist.

## Conclusion

The use of a 3D full body scanner produced results that strongly correlate to manually measured anthropometric measures. Predictions were improved substantially by including multiple measurements, which can only be obtained with a 3D body scanner, in the models.

## 1. Introduction

Over the last four decades obesity has nearly tripled worldwide and has reached the level of a global pandemic [1–4]. High fat mass, especially in the abdomen (visceral fat), and in connection with obesity, is associated with several diseases, such as coronary heart disease, diabetes mellitus type II and some types of cancer, as well as with all-cause mortality [5–12]. High muscle mass, on the other hand, appears to be beneficial for health and is associated with a reduced risk of functional impairment, disability and mortality, particularly later in life [13–15]. Since fat and muscle mass have such contrasting health implications, measurements of body composition are increasingly important in clinical practice as well as in medical research.

However, there is currently no universally suitable method to measure body composition. Each technique has advantages and disadvantages and its use is therefore highly situational. Standard imaging methods (DXA, CT and MRI) allow body composition to be assessed with high precision by distinguishing between fat and fat-free mass [16–19]. However, these techniques are time-consuming, expensive and/or invasive, and therefore inadequate for study settings in which many probands have to be examined with minimal health risks and in a short period of time [16,17,20]. Bioelectrical Impedance Analysis (BIA) is often used in such settings [16,21,22] because of the easy handling, high measuring speed and transportability of the measuring device. Nevertheless, BIA has some limitations. It is less precise than standard imaging methods [16,21,22], and provides numbers but no visualisation of fat tissue or any record of visible body characteristics. Thus, the results of BIA may seem abstract to lay people.

Despite their inaccuracy, classical manual anthropometric measurements are still most widely used both to estimate a person's fat mass and the associated health risk in epidemiological studies, and in clinical practice [23]. The most common anthropometric measurements associated with fat mass are Body Mass Index (BMI), Waist Circumference (WC) and Waist to Height Ratio (WHtR) [21]. All these measurements have been shown to be associated with body composition and are therefore able to predict fat mass [22,24,25]. These measurements only require minimal equipment and are based on visible body characteristics with intuitive meaning. They do, however, vary in their significance and precision in different population groups [26], and each of the measures has limitations. The BMI, for example, does not distinguish between fat mass and muscle mass/fat free mass and does not take account of fat distribution [22]. Measurements based on waist girth (WC or WHtR) have therefore gained popularity and seem to be better predictors for fat mass (particularly visceral fat) than the BMI [21,22,24,25,27]. However, these measurements and proportions provide no information on the composition of extremities, where the relative amounts of fat and muscle mass may vary greatly depending on a person's physical activity, or due to muscle loss in elderly people. Thus, changes in body composition cannot be assessed. Also, interobserver variability in anthropometric measurements may be an issue [23].

Variation in whole-body composition is probably too multidimensional to be properly measured by single distance measurements. As a new procedure to capture this multidimensionality, 3D photonic full body scans can be used [28–36]. Measurements are non-invasive,

safe, rapid and simple, and devices are transportable, although less easily than BIA devices. The 3D body scanner functions via photonic scanners surrounding the body. Within 12–15 seconds, millions of data points are gathered, creating a precise 3-dimensional body shape map with detailed body surface topography, from which about 150 standard measures can be derived automatically. These include circumferences of specific body parts, linear dimensions, cross-sectional areas, surfaces, segmental volumes, and proportions, all of which can be used, individually or in combination, to predict health-relevant parameters such as body composition [30,36–40]. Even the established simple predictors of body fat mentioned above (WC, WHR and WHtR) may be measured more rapidly and reliably using the body scanner than manual methods because the improved standardization reduces inter-observer variability [28,30,33,34,37–42].

However, some methodological questions still need to be addressed before 3D body scans can be implemented as a standard method for assessment of body composition. First, it is necessary to check whether manual and scanner-derived anthropometric measures are exactly comparable or whether they differ systematically, so that standard definitions of health risk classes (e.g. increased health risk with a WC greater than 94 cm [26]) must be adapted for scanner-derived measures. As regards the potential use of multiple measures for a more precise prediction of body composition, previous studies only considered a limited number of predefined measures [42], and only few of them used automatic variable selection procedures to identify the best predictors [43–45]. Because some of the 150 standard measurements are strongly correlated among each other, model selection procedures and other techniques such as 3D surface geometry may have to account for these correlations [46–51]. Still, further research is needed to identify which of the 150 standard measurements are most relevant for the prediction of body composition, or whether multiple (and partly strongly correlated) measurements are relevant, and how they should be selected or combined to obtain the most reliable predictions.

In this study we analysed a cross-sectional sample of 104 young men and asked the following research questions: Are standard anthropometric measurements assessed manually and by the scanner differently associated with body composition (fat and muscle mass) as estimated by BIA)? Are these predictions of body composition (as estimated by BIA) improved by considering additional measurements provided by the scanner? How many measurements should be included in a multivariable model to obtain the most precise predictions of body composition?

## 2. Material and methods

This study was part of a larger project in which we examined 104 recruits at the beginning of their Armed Forces basic training [52]. The cross-sectional baseline examinations involved young males (age range 18.8–24.4 years, mean 20.5 years, SD = 1.1 years) recruited by the Swiss Armed Forces and were conducted in Kloten (Canton of Zurich) from 21 March to 24 March 2017. Participation was voluntary and regarding socioeconomic status, origin or other demographic factors, and participation was voluntary. All young men were Swiss nationals (a precondition to be conscripted for mandatory Service for the Armed Forces), but information about migration background or ethnicity was not systematically collected in the questionnaire. Before beginning the study the participants were informed twice about its content and procedure, first in writing and then orally. In addition, informed consent was confirmed in the form of a signature. The ethics committee of the Canton of Zürich formally approved this study (No. 2016–01625).

Because in the setting of the presented study (limited time available for measurements within normal army operations) it was not feasible to perform invasive and more time-

consuming examinations, bioimpedance analysis was used to estimate whole body fat mass, visceral fat mass and skeletal muscle mass. The device used was a medical 8-point body composition analyzer (Seca mBCA 515, Seca AG, Reinach, Switzerland), which was validated in several studies and has often been used to compare body composition measures obtained through different measurement methods including 3D body scanners [53–61]. The participants stood barefoot on the four foot-electrodes and grasped the four hand-electrodes with their hands. Alongside the analysis of the body composition, selected anthropometric measurements which are relevant in a medical and epidemiological context were taken manually according to WHO guidelines [26]. Waist circumference (WC) was measured with a hand held-tape measure with stretch resistant quality and automatic retraction (Seca 201, Seca AG, Reinach, Switzerland). Participants were measured at the midpoint between the lowest point of the ribcage and the highest point of the pelvis bone, always by the same trained and experienced researcher. Height and weight were measured with a standard stadiometer (Seca 274, Seca AG, Reinach, Switzerland). The participants wore underwear and stood straight with their feet together.

For the 3D body scan a semi-mobile Anthroscan VITUSbodyscan body scanner was used. Four lasers and eight cameras create a point cloud via optical triangulation containing 300 data points per cm3. The software (Anthroscan 2016, Version 3.5.3) then calculates 150 standard measurements (ISO 7250 / ISO 8559 and DIN EN ISO 20685) including various girths and body part volumes. These body volume estimations (also for body regions) have been shown to be important for relative body fat mass [62] and of good validity and reliability in other studies [63,64]. For this study we included 90 standard measurements as delivered by the software (a complete list with measurement ID numbers is provided in S1 Table). The non-selected measurements were excluded before the start of data analysis based on two criteria: a) Specific measures intended for the textile sector (e.g. for shirts). b) Clearly redundant measurements (e.g. several nearly identical measures for leg length). The scanner was calibrated daily before use according to the manufacturer's instructions. The participants were scanned wearing tight-fitting underwear in standard position defined by the manufacturer of the 3D body scanner (standing up straight, feet positioned ca. 30 cm apart, arms slightly bent at the elbow and held slightly away from the body, head in accordance with the Frankfurt Horizontal Plane) and held breath after exhalation. To ensure the right positioning, we briefed every participant in advance. Participants wore form-fitting underpants and a tight-fitting bathing cap. Regarding postprocessing of the scans: We worked with the raw point clouds for the extraction of all standard measurementsexcept for the volumes. All 104 scans were checked for their quality (absence of artifacts). For the calculation of the partial volumes, we automatically calculated closed surfaces using the standard procedure in the Anthroscan software (good quality level, medium mesh size), the cutting off of the partial volumes was performed fully automatically via the Anthroscan software, but supervised for quality.

## Statistical methods

The agreement between manual and scanned anthropometric standard measurements was assessed through Bland-Altman plots, i.e. by plotting the difference between the two measurements against their mean value for each participant [65,66]. Smoothing lines in these plots showed whether one method yielded systematically higher values than the other, and whether this discrepancy affected the entire range of measured values or only part of the range.

The association between scanned anthropometric standard measurements (BMI, WC, WHtR) and body composition (absolute and relative fat mass, visceral fat mass, and skeletal muscle mass (AFM, RFM, Visc, SMM)) was assessed by Spearman rank correlations and

scatter plots with smoothing lines. These plots showed approximately linear relationships between absolute or relative fat mass and each of the predictors, and clearly segmented relationships for visceral fat mass, which was only linearly related to the measurements above a certain threshold. Accordingly, either linear regression or segmented regression was used to compare body fat predictions obtained with manual and scanned standard measurements by computing both the fraction of variation explained ($r^2$), and the prediction standard error, i.e. the square root of the mean squared prediction error obtained by leave-one-out cross-validation. We chose the cross-validation method over method of the splitting the data set in to training and validation data sets because of rather small overall sample size.

The possible gain in predictive value obtained by considering scanned anthropometric measurements other than the standard ones (BMI, WC, WHtR) was assessed by stepwise forward model selection using the AIC criterion. Of the three standard anthropometric measurements, only WC was considered here because it proved to be the best predictor for the three measures of body fat content. The first step of model selection showed whether any of the other 89 scanner measurements would predict body fat content better than WC. Further steps showed how much the prediction could be improved by adding a second, third or more predictors. For easy interpretation, the gain in predictive value was described as the fraction of variation in additional body fat content that was explained when a predictor entered the model. Because some of the 90 scanner measurements were strongly correlated with each other, we expected model selection to be partly arbitrary and determined by random structures in the data. To assess the resulting uncertainty in the choice of the best predictors, we repeated model selection for 2000 bootstrap samples of the data and recorded the first six predictors selected with each sample. We then determined how often individual measurements were selected in the first step, and how often each of the measurements initially selected among the six top predictors were also among these in the bootstrap samples.

Because stepwise model selection tends to overfit the data and produce unreliable solutions when predictors are strongly correlated, we also performed model selection with the lasso procedure. This involves fitting a multiple regression model with a penalized least squares criterion so that most of the unimportant and/or correlated predictors have a coefficient of zero and are dropped from the model. The optimal penalty term was selected by cross-validation using the "minimum + 1se" rule. We compared the predictions obtained with both model types in terms of explained variation ($r^2$), and prediction standard error from leave-one-out cross-validation.

Finally, we fitted a single multivariate lasso model to the four measures of body composition to obtain a single set of scanner measurements that would jointly provide the best predictions for the four body composition measures. We standardized both the scanner measurements and the four composition measures to a mean of 0 and standard deviation of 1 so that we could directly compare the regression coefficients and thus, the relative contribution of each of the selected scanner measurements to the prediction of each measure of body composition.

All analyses were performed using R version 3.5.2 (2018, The R Foundation for Statistical Computing, Vienna). To obtain the Bland-Altman plots we used blandr, the segmented regression was determined using the segmented package, and Lasso models we obtained using glmnet.

## 3. Results

The descriptive statistics for all manual and scanner measurements are reported in S1 Table. According to standard definitions of BMI categories, 20.2% of the participants were

overweight (BMI 25.0–29.9kg/m2) and 5.8% obese (BMI> = 30.0kg/m2). According to the WC, only 4.8% of the participants showed increased disease risk (WC 94-102cm) and 3.8% very high disease risk (WC>102cm), whereas the WHtR suggested that 17.3% had an increased disease risk (WHtR 0.5–0.6) and 1.0% a very high disease risk (WHtR>0.6). The three scanned anthropometric measurements (BMI, WC, WHtR) strongly and positively correlated to each other (Spearman Rho >0.89) (S1 Fig). Visceral fat mass, AFM and RFM were also strongly and positively correlated to each other (Spearman Rho >0.79), whereas the correlations with SMM were weaker (Rho 0.31–0.58) (S2 Fig).

In terms of agreement between methods, scanned and manually measured height, WC, WHtR, and BMI were strongly correlated (Spearman Rho>0.96) (Fig 1). However, the Bland-Altman plots for height showed a constant bias of -1cm towards scanned height being shorter, which resulted in slightly higher BMI values from the scanner. For WC and WHtR there was a trend towards values in the upper part of the range in the scanner than when manually measured.

The associations between the scanned anthropometric measurements for excess weight (BMI, WC, and WHtR) and visceral fat mass, AFM, RFM, and SMM are reported in Fig 2 and Table 1. In general, explained variation and prediction standard errors were similar between scanned and manual standard measurements. The highest explained variation ($r^2$) was observed for AFM and the lowest for SMM. Visceral fat mass showed segmented associations with all anthropometric standard measurements (breakpoint for WC = 78.4 cm). Overall, WC explained more variation than the two other anthropometric standard measurements.

Among the scan parameters, circumferential measurements in the abdominal and hip area were highly correlated with relative fat mass (Spearman Rank correlation Rho >0.8). Partial volumes had the highest correlations with skeletal muscle mass (Rho >0.8) (S1 Table). Vertical length and distance measurements showed generally showed weaker correlations with relative fat mass and skeletal muscle mass (Rho <0.4). As expected, predictors belonging to the same measurement type were positively correlated with each other:The average Spearman rank correlations (Rho) of predictors within groups were 0.71 forvertical distances, 0.73 for girths, and 0.79 for partial volumes. Associations among individual scan features are further illustrated by a tree from cluster analysis in S1 Fig.

Stepwise forward model selection confirmed that either WC or a closely related measurement (e.g. belly circumference or maximum belly circumference, high hip girth) was the single best predictor of body fat content (Table 2). In the bootstrap samples, WC was selected most often as a predictor of visceral fat mass, while belly circumference was selected most often as a predictor of AFM and RFM. The inclusion of a second predictor into the model increased the explained variation by 1.2% to 3.2%, and a third predictor explained a further 1.1% to 1.6%. Another 2.0% to 2.5% of variation was jointly explained by predictors 4 to 6. However, most of these predictors were selected among the top six predictors with fewer than 50% of the bootstrap samples, meaning that other measurements could be selected as well. Forearm volume (left or right) was most often selected as the best predictor of SMM, with various measures of leg size an alternative or second predictor, indicating that SMM was mainly related to total limb volume.

The total number of predictors selected by stepwise forward model selection ranged from 19 (AFM) to 49 (visceral fat mass). Both model fit (r-squared) and predictive value (cross-validated r-squared) increased or remained stable up to this large number of predictors (Fig 3). Model fit reached values close to 100%, and predictive value reached more than 90% for the four body composition measures. For the three measures, the mean prediction error of the stepwise selected model, as determined by cross-validation, was small and only slightly larger than the model's residual standard error despite the large number of predictors included

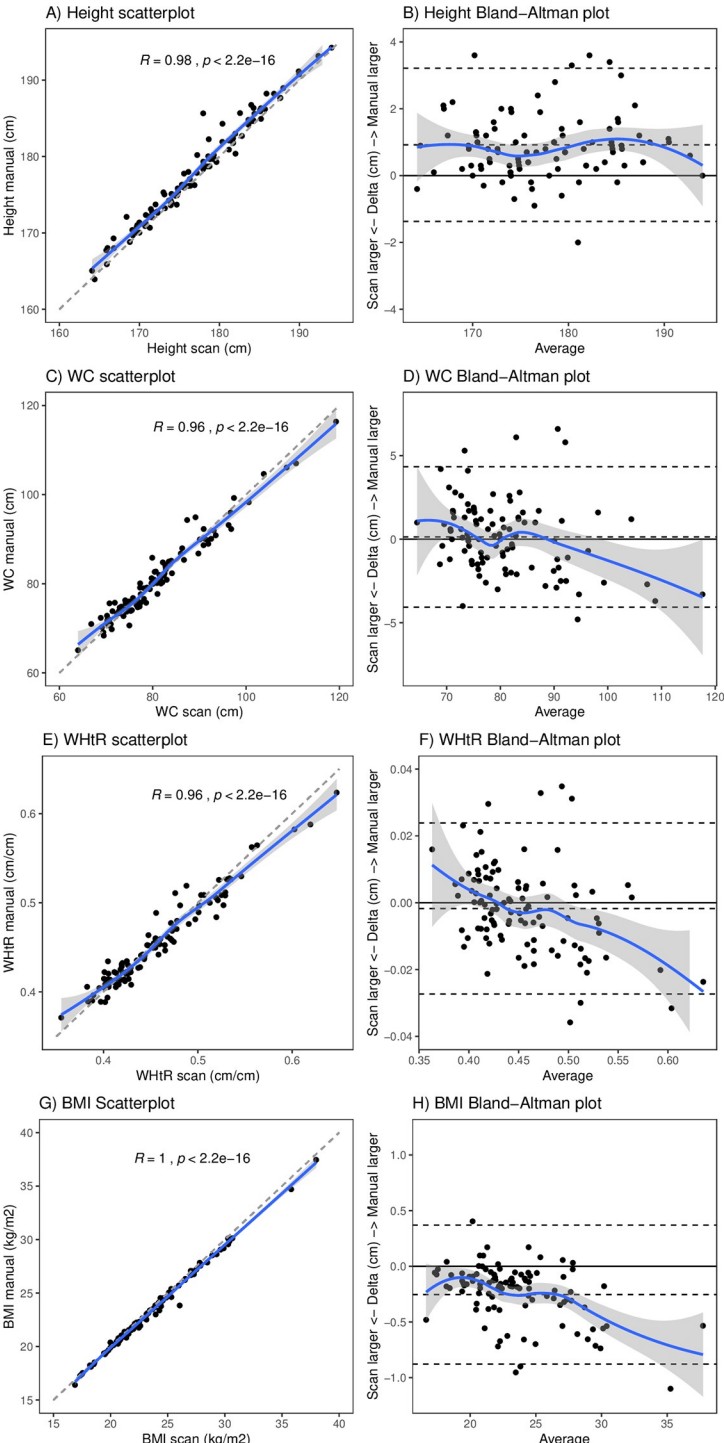

**Fig 1. Agreement between methods: Scan vs. manual by scatterplots (left) and Bland Altmann plots (right) for height (A,B), WC (C,D), WHtR (E,F) and BMI (G,H).** Generally, scanned and manually measured values are strongly correlated (Spearman Rho>0.96). For height there is a constant bias of -1cm towards scanned height being shorter. For WC and WHtR there is a trend towards higher values being larger in the in the scanner than when manually measured.

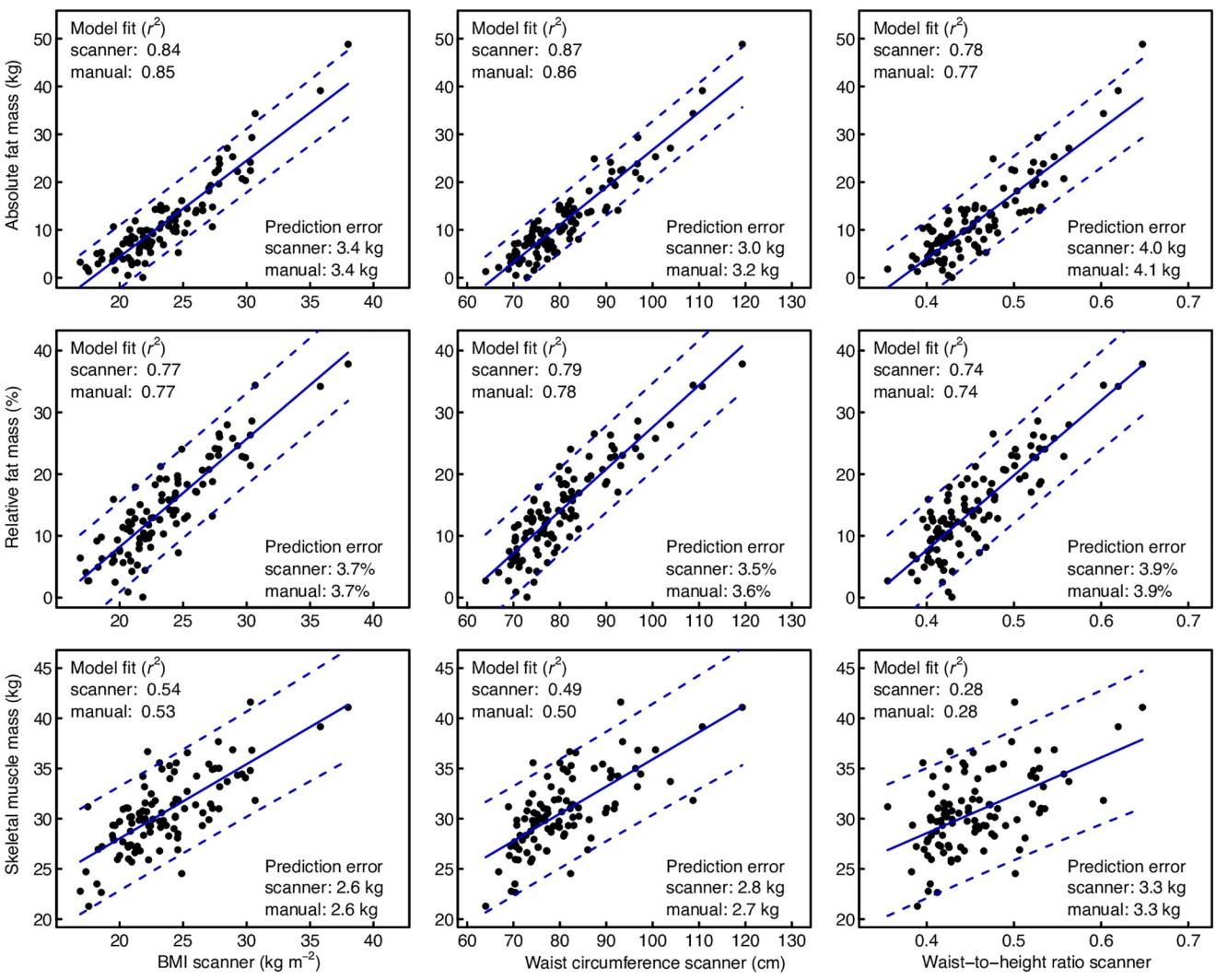

**Fig 2. Relationships between body composition (fat or muscle mass as determined through bioimpedance analysis) and three standard anthropometric measurements determined either with the 3D body scanner or through manual measurements.** Segmented regression was used for visceral fat, and linear regression for the other body composition measures. The fit of each regression model is given as explained variation ($r^2$), and its predictive value is given as the prediction standard error, i.e. the square root of the mean squared prediction error obtained by leave-one-out cross-validation.

(Table 1). The lasso procedure selected models with less predictors, ranging from 6 (Visc) to 19 (SMM), and with slightly lower predictive value (Table 1). Overall, we only found a moderate degree of overfitting (i.e., only small differences between r² and cross-validated r²) even with multiple predictors.

## 4. Discussion

In this study we compared estimated body composition (fat and muscle mass) with external body measurements obtained either manually or with a 3D body scanner in a cross-sectional sample of young Swiss men. We found that standard body measurements obtained with both methods were strongly correlated, yet some systematic differences existed. In general, standard measurements obtained with both methods performed equally well in predicting variation in

**Table 1. Comparison of univariable and multivariable regression models for the prediction of body composition (fat or muscle mass as determined through BIA) from anthropometric measurements.** In the univariable models, the four measures of body composition were related to three standard anthropometric measurements (BMI = body mass index, WC = waist circumference, WHtR = waist-to-height ratio, determined either with the 3D body scanner or through manual measurements) using linear regression or (for visceral fat) segmented regression. In the multivariable models, the four measures of body composition were related to 87 scanned measurements, from which the relevant predictors where selected either through stepwise forward model selection (to minimize the AIC) or through the lasso procedure. The fit of each model is given as explained variation ($r^2$), and its predictive value is given as the prediction standard error, i.e. the square root of the mean squared prediction error obtained by leave-one-out cross-validation.

| | Univariable models (standard measurements) | | | | | | Multivariable models | |
|---|---|---|---|---|---|---|---|---|
| | BMI | | WC | | WHtR | | Stepwise selected | Lasso |
| **Visceral fat (kg)** | scanner | Manual | scanner | manual | scanner | manual | p = 49 | p = 6 |
| Explained variation ($r^2$) | 0.76 | 0.76 | 0.87 | 0.92 | 0.83 | 0.88 | 0.987 | 0.834 |
| Prediction standard error (kg) | 0.48 | 0.47 | 0.36 | 0.27 | 0.43 | 0.35 | 0.250 | 0.429 |
| **Absolute fat mass (kg)** | | | | | | | p = 19 | p = 15 |
| Explained variation ($r^2$) | 0.84 | 0.85 | 0.87 | 0.86 | 0.78 | 0.77 | 0.978 | 0.947 |
| Prediction standard error (kg) | 3.38 | 3.36 | 3.01 | 3.19 | 4.03 | 4.13 | 1.638 | 2.435 |
| **Relative fat mass (%)** | | | | | | | p = 39 | p = 10 |
| Explained variation ($r^2$) | 0.77 | 0.77 | 0.79 | 0.78 | 0.74 | 0.74 | 0.975 | 0.888 |
| Prediction standard error (%) | 3.71 | 3.65 | 3.54 | 3.63 | 3.89 | 3.90 | 2.183 | 2.798 |
| **Skeletal muscle mass (kg)** | | | | | | | p = 23 | p = 19 |
| Explained variation ($r^2$) | 0.54 | 0.53 | 0.49 | 0.50 | 0.28 | 0.28 | 0.971 | 0.943 |
| Prediction standard error (kg) | 2.63 | 2.65 | 2.77 | 2.74 | 3.28 | 3.30 | 0.909 | 1.180 |

p = number of predictors selected.

estimated body composition. Of the 90 measurements obtained from the 3D body scans, the single best predictor of body fat was waist or belly circumference, while skeletal muscle mass was best predicted by limb size (length, girth or volume). The inclusion of additional measurements into multiple regression models increased the predictive value of each of the four body composition measures by more than 90%. Stepwise forward variable selection returned models with substantially more predictors than the lasso procedure, yet no overfitting was apparent, and a similar predictive value was achieved with both model selection procedures. However, due to strong correlations between some of the measurements, the exact choice of predictors in the best predictive model was largely arbitrary. Moreover, the optimal prediction function (possibly considering further derived features) still has to be determined in future studies.

The finding of a systematic bias between scanned and manually assessed data is supported by several other validation studies with similar results [42]. In the scanner participants stand with their legs hip-width apart (to enable the scanner/software to correctly identify the crotch), while they stand with their legs closer together when being manually measured [26]. In an earlier study with a different study population [40] as well as in the 54 follow-up assessments of the present study population [46] we found that height was systematically shorter in the scans. The positioning of the legs (hip-wide apart in the standard scan position vs. legs together and straight posture as being manually measured by an anthropometer) was found to be only partially responsible for the systematic height difference. The remaining difference could be related to the fact that we ran the automatic measurements with the software on the raw scans and point cloud might be slightly fragmented on the top of the head and the bottom of the feet. In other studies, height was systematically greater in the scans and that was mainly explained by issues related to the worn bathing cap (air beneath, or lots of hair up-biasing height) [34,42]. Future studies should therefore examine the partly conflicting results regarding the systematic height bias more closely, especially since the calculation of the BMI depends on it. Like in the 54 follow-up assessments of the present study population [46], WC was larger

**Table 2. Detailed results of stepwise forward model selection for the prediction of body composition (fat or muscle mass as estimated through bioimpedance analysis) from scanned anthropometric measurements.** For each of the six measurements selected first in the stepwise procedure, the (additional) fraction of variation in body fat or muscle mass explained by the inclusion of this predictor in the model is given. The stability of model selection was evaluated by running the procedure on 2000 bootstrap samples. For each of the six measurements initially selected first, the fraction of bootstrap samples where this measurement was also among the first six predictors selected is given. In addition, all measurements that were selected in the first step at least once are given.

| Variables | % expl. | Among first six (%) | Alternatives for the first (main) predictor (% of bootstrap samples where the variable was selected in the first step) |
|---|---|---|---|
| **Visceral fat (kg)** | | | |
| WC | 81.8 | 60.0 | WC (52.3), Belly circumference (18.3), High hip girth (13.1), Middle Hip (10.3), |
| Volume Forearm Right | 3.2 | 26.7 | Maximum belly circumference (5.2), High waist girth (0.85), Waist band (0.05) |
| Middle Hip | 1.6 | 32.0 | |
| Distance waistband knee | 0.8 | 17.8 | |
| Upper arm girth right | 0.7 | 20.4 | |
| Upper torso torsion | 0.5 | 11.3 | |
| **Absolute fat mass (kg)** | | | |
| Maximum belly circumference | 90.9 | 31.6 | Belly circumference (46.5), Maximum belly circumference (31.3), High hip girth (17.0), |
| Distance waist knee | 1.2 | 12.7 | Middle hip (4.6), WC (0.25), X_overview Volume (0.25), Buttock girth (0.05), Hip girth (0.05), Thigh girth right horizontal (0.05), Waist band (0.05) |
| X_Overview Volume | 1.1 | 56.0 | |
| Knee girth left | 1.1 | 32.9 | |
| Volume Forearm Left | 1 | 45.6 | |
| Forearm girth right | 0.5 | 8.8 | |
| **Relative fat mass (%)** | | | |
| Belly circumference | 83.3 | 88.5 | Belly circumference (88.5), Maximum belly circumference (9.5), High hip girth (1.75), |
| Thigh girth right horizontal | 3.1 | 35.5 | Thigh girth right horizontal (0.25), WC (0.2), Buttock girth (0.15), Hip girth (0.15), Thigh girth left horizontal (0.10) |
| Volume Forearm Left | 2.3 | 60.1 | |
| Dev. waist band from waist back | 1 | 11.3 | |
| min leg girth left | 0.8 | 24.3 | |
| Elbow girth right | 0.5 | 11.6 | |
| **Skeletal muscle mass (kg)** | | | |
| Volume Forearm Right | 78.5 | 63.2 | Volume forearm right (49.5), Volume forearm left (20.5), Volume lower Leg Right (8.6), X_Overview Volume (8.5), Volume Thigh Left (5.0), Forearm girth left (4.0), |
| Volume Thigh Left | 6.9 | 23.1 | Hip thigh girth (1.4), Volume Lower Leg Left (1.1), Total torso girth (0.85), Buttock |
| Waist to buttock height left | 2.7 | 9.9 | girth (0.15), Elbow girth right (0.15), min. leg girth left (0.15), calf girth right (0.10), Elbow girth left (0.05), Forearm girth right (0.05), min. leg girth right (0.05) |
| Neck height | 2.2 | 2.4 | |
| Forearm girth left | 1.8 | 42.6 | |
| Upper arm diameter left | 0.9 | 21.0 | |

in the scans, which is likely due to the tendency for hand-held tape measurements to compress the waist circumference, caused by too much tension, and thus reduce the values [64]. Moreover, the non-automatic positioning of the tape in manual measurements is challenging, even when conducted by trained personnel experienced in measuring overweight and obese people [67]. This might apply to an even greater degree for larger waist circumferences among obese people, based also on the findings of other studies which report low reliability of manual WC measurements in obese subjects [34,68].

In our study (regarding model fit) WC performed better than BMI and WHtR in the prediction of the three different body fat measures (visceral, absolute, relative). Moreover, the

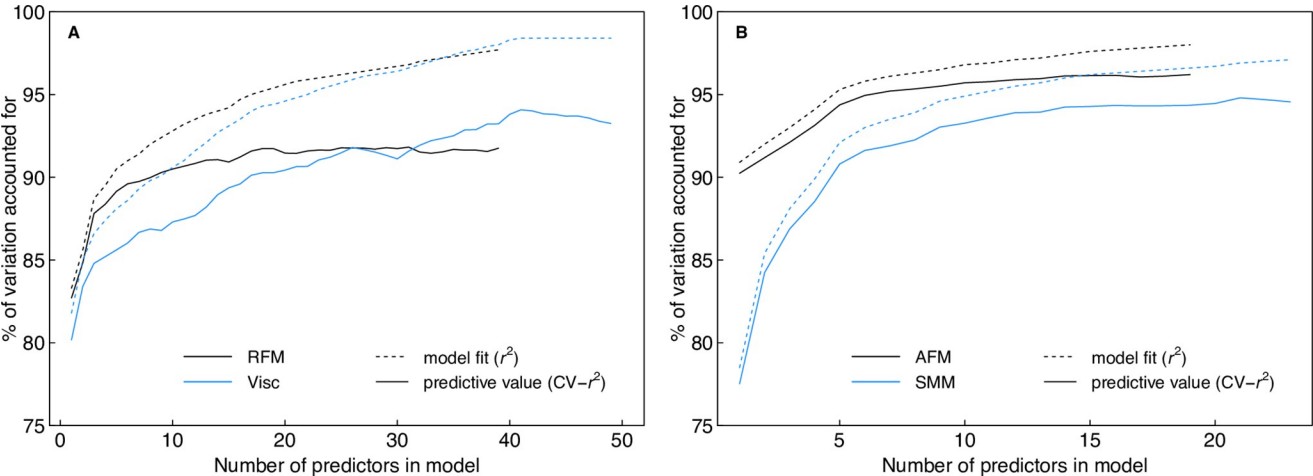

**Fig 3. Increase in model fit (r-squared) and predictive value (cross-validated r-squared) with increasing number of predictors in the model.**
Predictors were included in a stepwise forward selection procedure to minimize the AIC. Curves illustrate how many predictors were needed to obtain the maximal predictive value and the moderate degree of overfitting (difference between $r^2$ and cross-validated $r^2$) found even with multiple predictors.

association between the three manual anthropometric measurements with visceral fat mass was not linear, so we suggest not modeling this association with simple linear regression equations. In general, the univariable prediction of SMM performed lowest in our study. This is an indication that other body dimensions are more important in predicting muscle mass and that this prediction should be focused on the upper and lower extremities. Our findings support other studies which found only little differences between manually and scanned measures of waist circumference and their association with relative fat mass (as assessed by the same BIA device as in this study) [42]. While some studies have already shown a good validation of linear and circumferential measurements between 3D scans and manual anthropometry, there is not yet so much information on volume reconstructions. We provide here indications that these volumes have an added value, in our case especially in the estimation of skeletal muscle mass.

We furthermore support other studies that information gained from the scans results in high predictive values for body composition measures [41]. In our study we show that the multivariable prediction of body composition explains significantly more variability than the univariate prediction given by WC, BMI and WHtR. Thus, scanning the participants brings with it an additional benefit because within a short time a large number of additional body measurements are available, which markedly improves the prediction of body composition. Concerning the method of selecting individual variables from the whole catalogue of scanned measurements, there are not many other studies we can use for comparison. Previous studies have aggregated meta-measures to cluster body types [44,45] or have used deep learning [43]. Moreover, recent research has also focused on developing methods using the 3D geometry of the surface topography in order to predict body shape [51,69,70].

Strengths and limitations: One limitation of the present study is that body composition was estimated using BIA, which is not the gold standard. In a recent validation study [55], the Seca mBCA 515 BIA-device have been shown to be less reliable for visceral fat than other measures. However, in the setting of the presented study it was not feasible to perform invasive and more time-consuming examinations. We have only examined young Swiss men. However, the homogeneous sample also has advantages in that the precision of our results within the examination group is higher than that of a stratified sample. Also, our study was based on a relatively

small number or subjects. However, since the subjects of the present study regarding BMI and WC are closely comparable to the total population of all conscripts in Switzerland (which covers >90% of a given male birth cohort) [71,72], we believe that our results are generalizable, at least for young men in Switzerland. A larger and more diverse group of subjects will be needed to produce results which are generalizable for a broader population. Also, information about migration background and ethnicity (which influences body shape) should be collected in similar studies, and statistical methods validated in this study have to be tested in other data sets. Last but not least, the geometry accuracy of the mesh surface reconstruction performed by the Anthroscan VITUSbodyscan software has not yet been validated [51,69,70], which might be relevant to the volume estimations in our study.

## 5. Conclusion

Digital anthropometry is currently transforming areas of clinical nutrition assessment and provides new research opportunities [73]. We provide evidence, that even in smaller and homogenous samples prediction of body composition can be improved by making use of a broad range of standard scan measurements via various regression techniques. However, in order to make the best possible use of this technology, further studies that continue to lay important ground work must follow. The use of a 3D full body scan proved to be feasible for population based studies, and to produce results that strongly correlate to manually measured anthropometric measures. Some systematic differences remain and need further investigation. However, the use of a 3D scan might help in solving difficult situations like manual WC measurements in very obese individuals. As WC showed to be the best indicator for body fat in our study, this fact is of relevance for epidemiological applications. The 3D body scanner also allowed the prediction of skeletal muscle mass in our study. If this result is confirmed in other studies, and especially in a more varied population, this might reduce the use of multiple devices in population studies in the future.

## Supporting information

**S1 Table. The 90 selected measurements, including names, system-ID, mean and Standard Deviation (SD).** Rho RFM indicates Spearman rank correlations with RFM, and Rho SSM correlations with SMM (only Rho>0.5 are reported to provide an overview, and Rho>0.8 are reported in bold numbers).
(DOCX)

**S1 Fig. Correlation (Spearman) matrix for the three anthropometric measurements (BMI, WC, WHtR).**
(DOCX)

**S2 Fig. Correlation (Spearman) matrix for the four body composition measurements (visceral fat mass, AFM, RFM, SMM).**
(DOCX)

**S1 Data.**
(CSV)

## Acknowledgments

This paper was part of Roman Sager's medical Master thesis. The authors are especially thankful to Andreas Stettbacher (Chief Medical Surgeon), Franz Frey, Alexander Faas, Martino Ghilardi, Marco Müller, and Yvanka Jerkovic from the Swiss Armed Forces for their tremendous

logistical support. We also thank the IEM collaborators Nikola Koepke, Joël Floris, Lena Öhrström, Gülfirde Akgül, Anne Lehner, Lafi Aldakak, Michael Strässle, Patrick Eppenberger, Claudia Beckmann, and Nakita Frater for helping to collect the data. Furthermore, the authors thank Marcel Zwahlen, Ben Spycher and Jonathan Wells for helpful comments and discussions.

## Author Contributions

**Conceptualization:** Frank Rühli, Nicole Bender, Kaspar Staub.

**Data curation:** Roman Sager, Nicole Bender, Kaspar Staub.

**Formal analysis:** Sabine Güsewell, Kaspar Staub.

**Funding acquisition:** Frank Rühli.

**Investigation:** Roman Sager, Sabine Güsewell, Nicole Bender, Kaspar Staub.

**Methodology:** Sabine Güsewell, Kaspar Staub.

**Project administration:** Roman Sager, Frank Rühli, Nicole Bender, Kaspar Staub.

**Resources:** Frank Rühli, Kaspar Staub.

**Software:** Sabine Güsewell.

**Supervision:** Frank Rühli, Nicole Bender, Kaspar Staub.

**Visualization:** Sabine Güsewell, Kaspar Staub.

**Writing – original draft:** Roman Sager, Sabine Güsewell, Kaspar Staub.

**Writing – review & editing:** Sabine Güsewell, Frank Rühli, Nicole Bender.

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
