## [Decision Letter · Decision Letter 0]

2 Jan 2020

PONE-D-19-26892

Predicting body composition from 3D laser-based photonic body scans in young men

PLOS ONE

Dear Dr. Staub,

Thank you for submitting your manuscript to PLOS ONE. After careful consideration, we feel that it has merit but does not fully meet PLOS ONE’s publication criteria as it currently stands. Therefore, we invite you to submit a revised version of the manuscript that addresses the points raised during the review process.

The manuscript is well assessed by the two Reviewers; however, several major critiques are raised in the present form. Read carefully the Reviewers' comments and respond them appropriately.

We would appreciate receiving your revised manuscript by Feb 16 2020 11:59PM. To enhance the reproducibility of your results, we recommend that if applicable you deposit your laboratory protocols in protocols.io, where a protocol can be assigned its own identifier (DOI) such that it can be cited independently in the future. For instructions see: http://journals.plos.org/plosone/s/submission-guidelines#loc-laboratory-protocols

We look forward to receiving your revised manuscript.

Kind regards,

Masaki Mogi

Academic Editor

PLOS ONE

Journal Requirements:

3. Please remove your figures from within your manuscript file, leaving only the individual TIFF/EPS image files, uploaded separately.  These will be automatically included in the reviewers’ PDF.

Reviewers' comments:

Reviewer's Responses to Questions

**Comments to the Author**

1. Is the manuscript technically sound, and do the data support the conclusions?

Reviewer #1: Partly

Reviewer #2: Partly

2. Has the statistical analysis been performed appropriately and rigorously? 

Reviewer #1: No

Reviewer #2: I Don't Know

3. Have the authors made all data underlying the findings in their manuscript fully available?

Reviewer #1: No

Reviewer #2: Yes

4. Is the manuscript presented in an intelligible fashion and written in standard English?

Reviewer #1: Yes

Reviewer #2: Yes

5. Review Comments to the Author

Reviewer #1: The authors presents a paper comparing both manual anthropometric measurements and measurements derived from a 3D laser photonic scanner to BIA measurements. The paper is well written but overly complex and the statistics needs to be slightly revisited.

Main considerations:

(1) The model construction AND the validation is based on the same dataset, so the validation comparing measurements derived from a 3D laser photonic scanner to BIA measurements will of course give a high comparability. Since the model was optimized for exactly these datasets. In order to complete a more reliable validation the model parameters should not be selected in the same group as the BA plots and regression testing was performed in. Sugget to either split the group in two where the model parameters are determined in the first sub-group and the validation is performed in the second sub-group, or perhaps the follow-up 54 datasets could be used for the validation?

(2) The research questions are overly complex. There are six distinct research questions identified in the aims and these are strongly related. Is it necessary with such complex and many research question? It seems, in the paper, as the main questions rather are (i) to define model parameters for fitting 3D laser photonic images to body composition, and (ii) to validate this model?

Minor consideration:

(1) In the discussion it is noted that participants were standing with their legs hip-width apart for the scanning, whereas they were standing with legs close together for the manual measurements. This will introduce a bias towards lower heights in the scanning. I presume the length of the legs could have been determined from the scanning parameters and therefor (simply using the Pythagorean theorem) the heights could have been corrected for?

Reviewer #2: This paper presents a study of using 3D digital measurements to predict body composition. However, the major issues for this paper are: 

1) Vary lack of novelty. There are many existing papers conducted a similar study and this paper has no outstanding points comparing to the others.

2) Minimal contribution to the research in this area. The significance of this paper is unknown.

3) The experiment is not-well designed, using BIA as the ground truth is questionable. And the accuracy of the 3D scanner is not fully validated.

4) The dataset is too small and the diversity is limited. It is hard to guarantee the result can be generalized well.

A) Data Source

Dataset is very small. Age range and ethnicity diversity are limited. Statistics detail is not provided. And the justification for using such a small dataset is inadequate. 

Using BIA as the ground truth is questionable. The accuracy of BIA is pretty low. The author does not give enough justification for using BIA as the ground truth.

B) Method

The author evaluated the agreement between manual and scanned anthropometric standard measurements. But the anthropometric is only limited to BMI, WC, and WHtR.

The geometry accuracy and measurement readability of the 3D scanner is not evaluated and justified, especially when users take a 3D scan with their T-shirts on. More evidence should be provided for the data cleanness. 

How the feature selection is conducted (training set size, validation set size, and test set size) is not clear. I suggest the author use a separate dataset to test the performance of selected features.

The author states there is no overfitting but there is not enough justification (like experiments, data) for this.

The author states they collected 89 measurements, but does not provide enough detail about the 89 measurements. What are the measurements, how are they correlated, is there a better way to formulate the prediction function? Why only several of them are useful?

Data analysis is performed but the author did not dig into why some features are selected and why the others are irrelevant and what is the interrelation between the features. Other than feature extracted directly from the software, feature mapping may be explored to see if there are better features derived from the simple features. 

The result lacks significance. 

C) Other

Related work/background does not cover the latest research in body composition analysis using 3D geometry.

The manuscript is hard to read and bad-organized.

6. PLOS authors have the option to publish the peer review history of their article (what does this mean?). If published, this will include your full peer review and any attached files.

Reviewer #1: Yes: Janne West

Reviewer #2: No

---

## [Author Response · Author response to Decision Letter 0]

2 Apr 2020

A) Editor

“Thank you for submitting your manuscript to PLOS ONE. After careful consideration, we feel that it has merit but does not fully meet PLOS ONE’s publication criteria as it currently stands. Therefore, we invite you to submit a revised version of the manuscript that addresses the points raised during the review process.

The manuscript is well assessed by the two Reviewers; however, several major critiques are raised in the present form. Read carefully the Reviewers' comments and respond them appropriately.”

- Answer: Many thanks for giving us the opportunity to revise our manuscript. We are happy to comment on all points that the reviewers have noted. Please find our answers below.

B) Reviewer #1

“The authors presents a paper comparing both manual anthropometric measurements and measure-ments derived from a 3D laser photonic scanner to BIA measurements. The paper is well written but overly complex and the statistics needs to be slightly revisited.

Main considerations:

(1) The model construction AND the validation is based on the same dataset, so the validation compar-ing measurements derived from a 3D laser photonic scanner to BIA measurements will of course give a high comparability. Since the model was optimized for exactly these datasets. In order to complete a more reliable validation the model parameters should not be selected in the same group as the BA plots and regression testing was performed in. Sugget to either split the group in two where the model param-eters are determined in the first sub-group and the validation is performed in the second sub-group, or perhaps the follow-up 54 datasets could be used for the validation?”

- Answer: Thank you for this comment. As we wrote in the methods section of our article, we as-sessed the predictive value of our models using the standard method of leave-one-out cross-validation. The main reason for choosing this method was that our sample is relatively small. We have expanded the section on methodology to include this justification. The second reason was that our paper focuses on the potential benefit of including multiple predictors in a model, on the number of potentially relevant predictors and the way to select them, rather than the con-struction and testing of a single optimal predictive model. We used bootstrapping to assess the stability of predictor selection and showed that the identity of the predictors included in models varied substantially among bootstrap samples. We modified the title of the paper to clarify this purpose. The presentation of a single model with regression coefficients in Table 3 (multivari-ate lasso) may have suggested that this was proposed as optimal predictive model. In fact, Ta-ble 3 had a different purpose, which is why we did not validate this model. We agree that this was not sufficiently clear. To avoid increasing the complexity of the paper, we removed Table 3 and instead improved the description of individual predictors in the Supplementary Table 1. Finally, please note that the first part of the results directly compares manual and scanned measurement (BA and scatter plots) and does not involve any model construction or model pre-dictions. The measurements were chosen a-priori due to their established use in medical re-search and public health assessments. This is why no model validation was performed here.

“(2) The research questions are overly complex. There are six distinct research questions identified in the aims and these are strongly related. Is it necessary with such complex and many research question? It seems, in the paper, as the main questions rather are (i) to define model parameters for fitting 3D laser photonic images to body composition, and (ii) to validate this model?”

- Answer: We agree that the research questions were complex in the initially submitted manu-script version. We therefore reduced the number of research questions from six to three in the revised version. However, as explained above, our aim was not to define and validate a single model for the prediction of body composition, but rather to compare models based on two data sources (manual and scanner) as well as different approaches to predictor selection. Thus, we could not reduce the research questions as much as proposed above.

“Minor consideration:

(1) In the discussion it is noted that participants were standing with their legs hip-width apart for the scanning, whereas they were standing with legs close together for the manual measurements. This will introduce a bias towards lower heights in the scanning. I presume the length of the legs could have been determined from the scanning parameters and therefor (simply using the Pythagorean theorem) the heights could have been corrected for?”

- Answer: Thank you for this excellent suggestion. In fact, we addressed the systematic height dif-ference in a separate paper, which was just published in PeerJ (https://peerj.com/articles/8095/). In this sub-analysis we compared two scan positions in ca. 50 follow-up individuals (standard position with legs hip-wide apart vs. legs together and straight posture as being manually measured by an anthropometer). Our conclusion is that leg position is only partially responsible for the systematic height differences, the rest might be explained by additional factors such as slightly cut point clouds on the top of the head and at the bottom of the feet, as well as posture differences. As the PeerJ study has been published in the meantime, we complemented the discussion of the present study accordingly. 

C) Reviewer #2

“This paper presents a study of using 3D digital measurements to predict body composition. However, the major issues for this paper are: 

1) Vary lack of novelty. There are many existing papers conducted a similar study and this paper has no outstanding points comparing to the others.”

2) Minimal contribution to the research in this area. The significance of this paper is unknown.”

- Answer: Thank you for this comment. Indeed, the number of studies in the medi-cal/epidemiological 3D full body scan area has increased pleasingly in recent years. Neverthe-less, in our view, there is still a research gap in which methodologies can be used to extract the most and relevant information from the cloud of points in order to explore associations with disease risks etc. By following a new path (comparing different statistical techniques to select parameters) and by showing that even in a small sample we can improve the prediction of body composition using these techniques and the full battery of standard measurements, we believe that we are making a valuable contribution to the 3D full body scan community. We added a sentence highlighting our contribution to the field to the conclusion paragraph.

“3) The experiment is not-well designed, using BIA as the ground truth is questionable. And the accu-racy of the 3D scanner is not fully validated.”

- Answer: We admit that the paper's title may have suggested that we regard BIA measurements as "true" measures of body composition. This simplification was necessary to avoid an overly long title. We clearly state in the introduction and again in the limitations section that BIA is not the gold standard for determining body composition. As we also state in the limitations, BIA was simply the only applicable method in our study setting (limited time window available with-in the Armed Forces basic training and no possibility to use more invasive techniques). We would also like to note that even when still not interchangeable with DEXA the latest generation of BIA devices (as for example the SECA mBCA 515 used in our study) has made progress, as shown by independent validation studies (https://www.mdpi.com/2072-6643/10/10/1469). We pick up your point by adding at various places in the manuscript that our BIA measurements are an “estimate” of body composition. Regarding the accuracy of 3D scanners, our own and other studies have shown a high reliability and validity of linear measurements (lengths, cir-cumferences, etc.) in comparison to manual measurements (the repeatability is even increased against manual measurements due to the reduced intra-observer errors). This is also the case for the latest highly precise and accurate Anthroscan VITUSbodyscan device used in our study and also in large medical cohort studies in Germany (NAKO, Life). However, regarding vol-umes there are not yet many validations studies. We add to this research gap by showing that especially volumes contribute to the estimation of SMM. We supplemented this added value to the discussion. 

“4) The dataset is too small and the diversity is limited. It is hard to guarantee the result can be general-ized well.”

- Answer: We agree that our sample is relatively small and generalizability may be limited. We also emphasize this fact at various points in our manuscript. However, we believe that this is al-so one of the contributions of our study: We show with that even with a small sample from a homogeneous population, multiple measurements derived from the scanner contribute to the prediction of body composition. We also show (Table 2) that the identity of the selected predic-tors is variable. We believe that the statistical methods (parameter selection) that we point out are transferable and can provide other studies with valuable information. We agree that the presentation of a single model with regression coefficients in Table 3 (multivariate lasso) may have suggested that this was proposed as optimal predictive model. We fully agree that this model is specific to our study population and may be not valid for people of different age and ethnicity. As explained above (answer to reviewer 1), we removed Table 3 from the manuscript to avoid a misunderstanding on its purpose.

“A) Data Source

Dataset is very small. Age range and ethnicity diversity are limited. Statistics detail is not provided. And the justification for using such a small dataset is inadequate.”

- Answer: Thank you for this comment. For sample size and diversity, see our previous answer. Regarding statistical detail, we added some more information about age distribution and eth-nicity of our probands in the data description. Further information is provided in Appendix Table 1.

“Using BIA as the ground truth is questionable. The accuracy of BIA is pretty low. The author does not give enough justification for using BIA as the ground truth.”

- Answer: For general use of BIA, see our answer to your third comment above. We added a little more text in the methods section, why we had to use BIA.

B) Method

“The author evaluated the agreement between manual and scanned anthropometric standard measure-ments. But the anthropometric is only limited to BMI, WC, and WHtR.”

- Answer: Unfortunately, we only had a limited time window available for measurements within normal Armed Forces operations. We were therefore unable to collect any other manual an-thropometric mass than these standard measurements (BMI, WC, WHtR). And more time would indeed be required to take other high-quality measurements such as the circumferences of the thigh or forearm. However, we think that for most readers of our article, the comparison be-tween manual and scanned measurements is particularly interesting for the standard measure-ments we compare (BMI, WC, WHtR), which are often used in medical and epidemiological studies.

“The geometry accuracy and measurement readability of the 3D scanner is not evaluated and justified, especially when users take a 3D scan with their T-shirts on. More evidence should be provided for the data cleanness.”

- Answer: Thank you for this important note that we were not precise enough about our data preparing process. We worked with the raw point clouds for the extraction of all standard measurements (but not the volumes). For the standard measurements, there was no postpro-cessing of the scans, only all 104 scans were checked for their quality (artefacts, etc.). For the calculation of the partial volumes, we automatically calculated closed surfaces using the stand-ard procedure in the Anthroscan software (good quality level, medium mesh size), the cutting off of the partial volumes was also done fully automatically via the Anthroscan software, but was supervised by us. We added corresponding text to the methods section. Regarding geomet-ric accuracy (which mostly applies for postprocessed scans (mesh)): We agree that, even with the high-precision Anthroscan VITUSbodyscan devices, there has not yet been a study that has validated the accuracy of the overlying mesh. However, the resolution of these devices is so high (even tattoos are precisely recorded, and these devices are also used in dermatology) that we are confident about their accuracy. However, this is less relevant in the context of the present study because we automatically extract most of the measurements from the raw point cloud (over 7 million points) and no proband wore a T-shirt during the measurements. However, we will in-clude this important point when we work with 3D surface geometry in future studies. And, we have added a corresponding sentence to the limitations section.

“How the feature selection is conducted (training set size, validation set size, and test set size) is not clear. I suggest the author use a separate dataset to test the performance of selected features.”

- Answer: Thank you very much for this comment. Please see our answer to the equivalent first comment of Reviewer 1. 

“The author states there is no overfitting but there is not enough justification (like experiments, data) for this.

The author states they collected 89 measurements, but does not provide enough detail about the 89 measurements. What are the measurements, how are they correlated, is there a better way to formulate the prediction function? Why only several of them are useful?”

- Answer: Regarding overfitting, we present evidence in Figure 3. In this visualization, curves il-lustrate how many predictors were needed to obtain the maximal predictive value and the mod-erate degree of overfitting (difference between r2 and cross-validated r2) found even with multi-ple predictors. We improved the corresponding part in the results section in order to better high-light the overfitting question. Regarding details on the measurements: We agree that some of these measurement designations are somewhat burdensome. But please note that we provide the exact system name and official ID numbers of all measurements used (based on standards ISO 7250 / ISO 8559 and DIN EN ISO 20685) together with descriptive statistics in Appendix Table 1. This should also facilitate comparison across studies with the same scanner device and soft-ware. Regarding correlations, we already noticed in our manuscript that some of these meas-urements are highly correlated. We added information on this to the main text as well as to supplementary Table 1, and we newly added Supplementary Figure 1. Regarding the prediction formula: As noted above, the aim of our paper was not to provide a single optimal prediction function, and we removed Table 3 to avoid this misunderstanding. We also state more clearly in the discussion that the optimal prediction function (possibly considering further derived fea-tures) still has to be determined.

“Data analysis is performed but the author did not dig into why some features are selected and why the others are irrelevant and what is the interrelation between the features. Other than feature extracted di-rectly from the software, feature mapping may be explored to see if there are better features derived from the simple features.”

- Answer: Thank you for this good comment. We illustrated the relationships among features in a new Supplementary figure (tree from cluster analysis). Furthermore, we classified the features into groups (height, girth, volume etc.) and evaluated correlations within and between groups, as well as their correlations with body composition (addition to Appendix Table 1). We could have made similar considerations for the multivariable models, but this was beyond the scope of the paper and would have made it even more complex. While other studies have attempted to make sense of the standard measurement batterie using PCA and other methods, our goal has been to work relatively close to the standard measures that the Anthroscan software provides. Since we have already used 90 measures for our predictions within 104 subjects, additional or combined measures would most probably caused overfitting issues. 

“The result lacks significance.”

- Answer: Please see our answer to your first comment.

C) Other

“Related work/background does not cover the latest research in body composition analysis using 3D geometry.”

- Answer: We were happy to take up this interesting new direction in the discussion.

“The manuscript is hard to read and bad-organized.”

- Answer: Thank you for this comment. We are aware that our manuscript is relatively complex. We have reduced the number of research questions (based on a comment from Reviewer 1), but it is in the nature of these remaining questions that their answers are and remain technical and therefore complex. Nevertheless, we have tried to reduce the complexity here and there and to offer more reader guidance.

---

## [Decision Letter · Decision Letter 1]

21 Apr 2020

PONE-D-19-26892R1

Multiple measures derived from 3D photonic body scans improve predictions of fat and muscle mass in young Swiss men

PLOS ONE

Dear Dr. Staub,

Thank you for submitting your manuscript to PLOS ONE. After careful consideration, we feel that it has merit but does not fully meet PLOS ONE’s publication criteria as it currently stands. Therefore, we invite you to submit a revised version of the manuscript that addresses the points raised during the review process.

Major revisions are still needed in the present form. See the Reviewers' comments and respond them appropriately.

We would appreciate receiving your revised manuscript by Jun 05 2020 11:59PM. To enhance the reproducibility of your results, we recommend that if applicable you deposit your laboratory protocols in protocols.io, where a protocol can be assigned its own identifier (DOI) such that it can be cited independently in the future. For instructions see: http://journals.plos.org/plosone/s/submission-guidelines#loc-laboratory-protocols

We look forward to receiving your revised manuscript.

Kind regards,

Masaki Mogi

Academic Editor

PLOS ONE

Reviewers' comments:

Reviewer's Responses to Questions

**Comments to the Author**

1. If the authors have adequately addressed your comments raised in a previous round of review and you feel that this manuscript is now acceptable for publication, you may indicate that here to bypass the “Comments to the Author” section, enter your conflict of interest statement in the “Confidential to Editor” section, and submit your "Accept" recommendation.

Reviewer #1: All comments have been addressed

Reviewer #2: (No Response)

2. Is the manuscript technically sound, and do the data support the conclusions?

Reviewer #1: (No Response)

Reviewer #2: Partly

3. Has the statistical analysis been performed appropriately and rigorously? 

Reviewer #1: (No Response)

Reviewer #2: N/A

4. Have the authors made all data underlying the findings in their manuscript fully available?

Reviewer #1: (No Response)

Reviewer #2: (No Response)

5. Is the manuscript presented in an intelligible fashion and written in standard English?

Reviewer #1: (No Response)

Reviewer #2: Yes

6. Review Comments to the Author

Reviewer #1: (No Response)

Reviewer #2: I have the same concern with the Reviewer #1 on how the model is evaluated. And I totally agree with Reviewer#1 that the author should separate the dataset, i.e., train the model using one part and do the evaluation using the other. At lease I think the author should give that result as part of the analysis.

7. PLOS authors have the option to publish the peer review history of their article (what does this mean?). If published, this will include your full peer review and any attached files.

Reviewer #1: Yes: Janne West

Reviewer #2: No

---

## [Author Response · Author response to Decision Letter 1]

22 Apr 2020

“I have the same concern with the Reviewer #1 on how the model is evaluated. And I totally agree with Reviewer#1 that the author should separate the dataset, i.e., train the model using one part and do the evaluation using the other. At lease I think the author should give that result as part of the analysis.”

Answer: Thank you very much for insisting on this really important point. We agree that models should be trained and tested on different data. We would like to stress again that this is exactly what we did when we applied leave-one-out cross-validation: We trained a model on (n-1) sub-jects and tested it on the remaining (excluded) subject, and repeated this in turn for all subjects in the sample. This procedure avoided the arbitrary subdivision of our rather small sample into even smaller subsamples. Please let us mention that Reviewer #1, who initially suggested the da-taset splitting, agreed with our explanation and with the use of this procedure for model evalua-tion given the aims of our study. We therefore believe that we have already sufficiently ad-dressed this Reviewer's concern and hope that we could clarify this here.

---

## [Editor Report · Decision Letter 2]

29 May 2020

Multiple measures derived from 3D photonic body scans improve predictions of fat and muscle mass in young Swiss men

PONE-D-19-26892R2

Dear Dr. Staub,

We are pleased to inform you that your manuscript has been judged scientifically suitable for publication and will be formally accepted for publication once it complies with all outstanding technical requirements.

With kind regards,

Masaki Mogi

Academic Editor

PLOS ONE

Additional Editor Comments (optional):

The authors responded to the Reviewer's comments.
---

## [Editor Report · Acceptance letter]

2 Jun 2020

PONE-D-19-26892R2 

Multiple measures derived from 3D photonic body scans improve predictions of fat and muscle mass in young Swiss men 

Dear Dr. Staub:

I'm pleased to inform you that your manuscript has been deemed suitable for publication in PLOS ONE. Congratulations! Your manuscript is now with our production department. 

Kind regards, 

on behalf of

Dr. Masaki Mogi 

Academic Editor

PLOS ONE